# Domain-Adaptive Syntax Tree Repair via Cross-Corpus Transfer with Adversarially Aligned Transformers

## Abstract

We propose a domain-adaptive syntax tree repair system that meets the challenges of code correction tasks of cross corpus generalization. The natural heterogeneity of code corpora in terms of domains biases the average algorithmic repair model most of the time to the extent that the performance is not optimal when applied to see programming contexts. To reduce this, we propose Domain-Aligned Syntax Tree Transformer (DASTT), a hierarchical neural model that simultaneously optimizes syntactic feasibility and domain-invariant features. The model takes raw source code as input through a byte pair encoding tokenizer and uses a multi-layer encoder of Transformer with adversarial training to align pairwise distributions of the tokens across domains. A gradient reversal layer reduces domain discrimination while maintaining the accuracy of repairs so that the system adapts to different codebases without ever needing to retrain. Furthermore, the decoder includes a pointer amplified mechanism to directly manipulate the syntax trees, inducing exact repair actions (insertion of nodes or deletion of nodes). The proposed method fits smoothly into the existing compiler pipelines, where existing lexers and parsers are substituted; compatibility with downstream activities is assured. Experiments show that DASTT outperforms domain-specific baselines on cross-corpus repair tasks by a large margin, achieving strong performance on multiple programming languages and coding styles. The adversarial alignment framework guarantees the syntactic fidelity even under large domain shifts and hence is suitable for real-world deployment in heterogeneous development environment. This work significantly advances the state-of-the-art on automated code repair by bringing together techniques of domain adaptation and structural syntax tree manipulation.

## 1 Introduction

The growing complexity of software systems has made automated code repair an essential tool for ensuring the quality of the software and productivity of the developer. Traditional approaches to syntax error correction often rely on handcrafted rules or domain-specific parsers, which struggle to generalize across diverse programming contexts (Fan et al., 2023). While recent advances in deep learning have shown promise for code repair tasks, these methods frequently exhibit performance degradation when applied to code from domains not well-represented in their training data (Zhang et al., 2023). This limitation is related to a general bias in code corpora, in which different programming paradigms (e.g., embedded systems vs. scientific computing) display different syntactic and stylistic patterns that test the capability of traditional repair models.

Existing work in the field of program repair has examined various types of popular representations of the code, ranging from raw form-text and abstract syntax trees (ASTs), with mixed success. Some approaches focus on learning embeddings from token sequences (Tian et al., 2020), while others leverage structural information from ASTs (Li et al., 2020). However, often these approaches make certain assumptions about the distributions of the training and test data, an assumption that is rarely true in a production environment. For instance, a model developed to learn about web development code might have poor performance when applied to low-level systems programming, because of the differences in its coding conventions and API usage. This domain gap restricts the practical use of

automated repair tools in real-world applications because codebases contain multiple languages and paradigms.

To handle these issues, we propose a domain adaptive transfer learning framework for syntax tree repair. Unlike previous research, which addresses code repair as a monolithic task, we provide a way to explicitly consider domain shifts by performing a shift between token and structural representations across different code corpora. The core innovation behind this work is that adversarial domain adaptation is combined with neural modeling extended with syntax awareness which can then learn strategies to repair which generalize across training domains. Specifically, we employ a masked language modeling objective during pre-training to capture syntactic regularities (Wettig et al., 2022), followed by adversarial training to minimize domain-specific biases in the learned representations (Tzeng et al., 2017). This is a dual optimization to make sure the model is both preservation-invariant (considering that it keeps its repair capabilities) and difference-invariant (as it doesn't consider superficial differences between code domains).

The proposed method has several advantages over the existing techniques. First, it removes the necessity for domain-specific tuning by automatically adapting to new programming contexts by adversarial tuning. Second, it leverages cross-corpus knowledge to improve repair accuracy, even for rare or domain-specific syntax errors (Tian et al., 2023). Third, the integration of byte-pair encoding (BPE) allows the model to handle out-of-vocabulary tokens, a common issue in code repair tasks (Araabi et al., 2022). These features render the system especially appropriate for use in heterogeneous development environments, where codebases tend to mix multiple languages and styles together.

Our contribution can be summarized as follows:

1. We propose a domain-adaptive syntax tree repair algorithm that employs transfer learning in conjunction with adversarial domain alignment, allowing marijuana performance across diverse code corpora.

2. We show that adversarial training was able to successfully or at least ameliorate the domain bias in code representations while retaining the ability of the model to carry out accurate manipulations of syntax trees.

3. We demonstrate empirically that the proposed method outperforms domain-specific baselines on cross-corpus repair tasks with state-of-the-art results on multiple programming languages.

The rest of this paper is organized as follows: Section 2 presents related work in related repair and domain adaptation. Section 3 gives some background on syntax tree representations and adversarial training. Section 4 presents the detailed information about the proposed framework including architecture and training objectives. Experimental results are provided in Section 5 and implications and future directions are discussed in Section 6.

## 2 RELATED WORK

Automated program repair has developed massively due to advance in machine learning and techniques for code representation. Existing approaches can be broadly divided into three different paradigms: rule-based systems, statistical machine learning models, and deep neural networks.

### 2.1 CODE REPRESENTATION LEARNING

The effectiveness of automated repair systems highly depends on the way of source code representation. Traditional methods often use handcrafted features or syntactic templates (Zhang et al., 2023), which struggle to capture the semantic nuances required for accurate repairs. Recent work has shifted toward learned representations, with several studies demonstrating the advantages of neural embeddings over manual feature engineering (Tian et al., 2020). These approaches generally use sequence-based models for processing raw sequence of code tokens or tree-based models for processing the structural information in the form of ASTs. For instance, (Li et al., 2020) uses RNNs to encode method-level code changes, while (Namavar et al., 2022) systematically compares various code representations for repair tasks.

## 2.2 TRANSFER LEARNING FOR CODE

Transfer learning has emerged as a powerful technique for adapting models across different programming domains. (Li, 2021) demonstrates how attention mechanisms and masked language modeling can facilitate knowledge transfer between programming languages. Similarly, (Mastropaolo et al., 2022) shows that subword units like byte-pair encoding help mitigate vocabulary mismatches across domains.

## 2.3 DOMAIN ADAPTATION TECHNIQUES

Domain adaptation techniques try to shut down distributional changes between source and target domains. In the context of program repair, (Bukhsh et al., 2021) explores both in-domain and cross-domain transfer strategies, highlighting the challenges of adapting repair models to new environments. Adversarial training has proven particularly effective for domain alignment, as demonstrated by (Zhang et al., 2025), which combines transfer learning with self-attention mechanisms for fault localization.

## 2.4 NEURAL PROGRAM REPAIR

Recent neural approaches to program repair have achieved promising results by leveraging large-scale pre-training and sophisticated architectures. (Jiang et al., 2021) frames repair as a neural machine translation problem, while (Jiang et al., 2023) investigates how encoder-only models can support repair tasks through code representation learning.

The proposed DASTT framework is different from current approaches in a number of key aspects. First, it directly treats domain shift in adversarial alignment, unlike typical repair systems based on the assumption of domain homogeneity. Second, it integrates structural and lexical information in a common space of representation, taking over the limitation of purely token-based or tree-based approaches. Third, the combination of gradient reversal allows to achieve feature learning which is domain invariant without sacrificing repair accuracy.

# 3 BACKGROUND AND PRELIMINARIES

To set up the technical foundation for our domain adaptive syntax tree repair framework, this section introduces 3 key concepts: syntax tree representations, byte pair encoding and Transformer architecture.

## 3.1 SYNTAX TREE BASICS

Syntax trees are a representation of program structure in a formal way by showing the hierarchical relations between the structuring elements of a program. In graph theory terms, we can model a syntax tree as a directed acyclic graph $G = (V, E)$, where $V$ denotes the set of nodes representing language constructs (e.g., statements, expressions) and $E$ represents edges indicating syntactic relationships (e.g., parent-child dependencies). Each node $v \in V$ is typically labeled with its syntactic category (e.g., "IfStatement", "VariableDeclaration"), while edges encode the compositional structure of the program (Neamtiu et al., 2005).

The tree structure inherently captures the context sensitive way that programming languages are written, that the meaning of an element of code can often depend on its position in the hierarchy. This property makes syntax trees particularly suitable for repair tasks, as they preserve both the lexical content and the structural constraints necessary for generating valid fixes (Si et al., 2019).

## 3.2 BYTE-PAIR ENCODING (BPE)

Byte-Pair Encoding tackles the vocabulary mismatch issue in the processing of neural codes by breaking the text into subwords that compose rare tokens or unseen tokens. The algorithm iteratively merges the most frequent pairs of bytes or characters, creating a vocabulary that balances expressiveness and generalization (Sennrich et al., 2015). Given a corpus of source code files, the

BPE process splits each token in the input into individual characters, and then performs merge operations based on co-occurrence statistics.

For programming languages, I find this method to be especially useful because there is a lot of shared subword information in tokens (for example, "getValue" and "setValue" have the suffix "Value"). The resulting subword vocabulary enables the model to handle out-of-vocabulary tokens that frequently appear in cross-domain scenarios, such as project-specific identifiers or library APIs not seen during training (Lakomkin et al., 2020).

### 3.3 Transformer architecture

At its core, the model is based on the multi-head self-attention mechanisms that compute dynamic representations by attending to all positions in the input sequence concurrently. A formula for the attention function for each head is:

$$\text{head}_i = \text{Attention}(\mathbf{Q}\mathbf{W}_i^Q, \mathbf{K}\mathbf{W}_i^K, \mathbf{V}\mathbf{W}_i^V) \tag{1}$$

where $\mathbf{Q}, \mathbf{K}, \mathbf{V}$ represent queries, keys, and values respectively, and $\mathbf{W}_i^Q, \mathbf{W}_i^K, \mathbf{W}_i^V$ are learned projection matrices for the $i$-th attention head (Vaswani et al., 2017). The complete multi-head attention combines these individual heads through concatenation and linear projection:

$$\text{MultiHead}(\mathbf{Q}, \mathbf{K}, \mathbf{V}) = \text{Concat}(\text{head}_1, ..., \text{head}_h)\mathbf{W}^O \tag{2}$$

This is naturally constructive architecture for both sequential and structural code representations. When processing syntax trees, the model can attend to parent nodes while generating repairs for children, maintaining the hierarchical constraints essential for producing syntactically valid fixes (Tang et al., 2022).

## 4 Domain-adaptive transfer learning for syntax tree repair

The proposed Domain-Aligned Syntax Tree Transformer (DASTT) framework encompasses various new components in order to achieve robust cross-domain syntax repair.

### 4.1 Adversarial domain alignment for code representations

From the above, the underlying problem of cross-domain syntax repair is finding the balance between the feature distributions of various code bodies and ensuring repair accuracy.

The coordinate-adversarial alignment is obtained by using a Gradient Reversal Layer (GRL) which is placed between the shared encoder and a domain classifier $D$. During forward propagation the GRL behaves as an identity function and the domain classifier then makes predictions on the source domain of the encoded features.

The complete adversarial loss function combines the standard cross-entropy repair loss $\mathcal{L}_{\text{repair}}$ with the reversed domain classification loss $\mathcal{L}_{\text{domain}}$:

$$\mathcal{L}_{\text{total}} = \mathcal{L}_{\text{repair}}(E, R) - \lambda\mathcal{L}_{\text{domain}}(E, D) \tag{3}$$

where $R$ denotes the repair decoder, and $\lambda$ controls the trade-off between domain invariance and repair accuracy. The domain classifier architecture is similar to regular Transformer layers but has a binaryout head and the repair decoder has additional pointer mechanisms for manipulating the trees.

### 4.2 Pointer-augmented tree editing mechanism

At each decoding step $t$, the model computes both a vocabulary distribution $p_{\text{vocab}}$ over possible output tokens and a pointer distribution $p_{\text{ptr}}$ over input nodes:

$$p_{\text{ptr}}(y_t = \text{node}_i) = \text{Softmax}(\mathbf{h}_t^{\text{dec}}\mathbf{W}_p\mathbf{h}_i^{\text{enc}}) \tag{4}$$

where $\mathbf{W}_p$ is a learned projection matrix, $\mathbf{h}_t^{\text{dec}}$ is the current decoder state, and $\mathbf{h}_i^{\text{enc}}$ represents the encoded input node features. The final output distribution interpolates between these two modes

using a learned generation probability $p_{\text{gen}} \in [0, 1]$:

$$p(y_t) = p_{\text{gen}}p_{\text{vocab}}(y_t) + (1 - p_{\text{gen}}) \sum_{i:x_i=y_t} p_{\text{ptr}}(i) \tag{5}$$

This mechanism allows the model to either grow new tokens, or directly copy nodes from the input tree, which allows for fine-grained control of the modifications made to the tree.

### 4.3 BPE-AUGMENTED SYNTAX TREE PROCESSING

In order to deal with the vocabulary mismatch across the domains, DASTT uses byte pair encoding at the token and structural level: For syntax tree nodes, we adapt this methodology by using learned embeddings for encoding node types and structural positions:

$$\mathbf{h}_i^0 = \mathbf{E}_{\text{type}}(t_i) + \mathbf{E}_{\text{pos}}(p_i) + \mathbf{E}_{\text{subword}}(s_i) \tag{6}$$

where $t_i$ denotes the node type (e.g., "IfStatement"), $p_i$ represents its position in the tree, and $s_i$ is the sequence of subword units for its textual content.

### 4.4 UNIFIED PRE-TRAINING AND FINE-TUNING STRATEGY

The pre-training phase uses a domain-adversarial variant of MLM where some tokens are masked not only for prediction but also for domain classification:

$$\mathcal{L}_{\text{pretrain}} = \mathbb{E}[\log p(x_{\text{masked}}|x_{\text{observed}})] - \lambda\mathbb{E}[\log p(d|h_{\text{masked}})] \tag{7}$$

This compels the model to form representations which are predictive of the masked tokens, but non-predictive of where the tokens were formed from during domain creation. During fine-tuning, we use the following to initialize such features: Domain-invariant. and optimize the combination of the repair and alignment goals from Equation 4: On gradual transition between a general pre-training phase and a task-specific fine-tuning phase, the model can make use of the cross-corpus knowledge while adapting to the very specific requirements of syntax repair.

### 4.5 END-TO-END PROCESSING PIPELINE

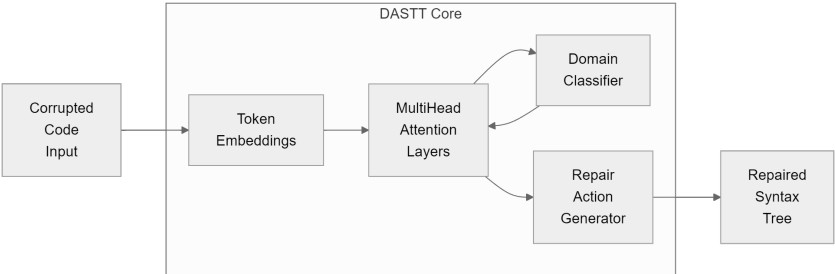

Figure 1: Internal Structure of DASTT. The framework processes raw code through BPE tokenization, adversarial encoding, and pointer-augmented decoding.

As shown in Figure 1, DASTT replaces the traditional lexer/parser pipeline with a unified neural architecture that processes raw code directly into repaired syntax trees. The input code undergoes BPE tokenization initial embedding look-up followed by passing it through the shared encoder with adversarial training.

The encoder stack is comprised of typical Transformer layers along with relative position encoding in order to learn both sequential and hierarchical relations:

$$\text{Attention}(Q, K, V) = \text{Softmax}\left(\frac{QK^T + R}{\sqrt{d_k}}\right) V \tag{8}$$

where $R$ includes learned relative position biases to assist in modeling tree parent-child distances. The decoder adds another cross-attention mechanism over encoder states with structural masks to valid tree transformations added.

## 5 EXPERIMENTAL EVALUATION

In order to verify the effectiveness of the Domain-Aligned Syntax Tree Transformer (DASTT) proposed in this work, we conducted extensive experiments across a range of programming languages and domains of code.

### 5.1 EXPERIMENTAL SETUP

**Datasets and Preprocessing**

We evaluated DASTT on a representative emasculation of code corpus in five languages (Python, Java, C++, JavaScript, and Go) and three application domains (web, scientific, and embedded systems). The datasets were constructed by parsing GitHub repositories using tree-sitter parsers (Latif et al., 2023), then extracting syntactically valid code snippets as positive examples.

**Baselines**

We have compared DASTT against 3 categories of baseline methods:

1. **Domain-Specific Models**: Separate Transformer models trained independently on each domain (Python-web, Java-scientific, etc.) (Kelly & Tolvanen, 2008)

2. **Conventional Repair Tools**: Rule-based systems including PMD (Singh et al., 2017) and Error-Prone (Tomassi, 2018)

3. **General-Purpose Neural Models**: CodeBERT (Feng et al., 2020) and GraphCodeBERT (Guo et al., 2020) fine-tuned on the repair task

All baselines of neural models were with comparable parameter numbers ( 150M), and they were trained with identical hardware resource usage. For fair comparison, we implemented the domain-adversarial versions of the CodeBERT and GraphCodeBERT using the same GRL setup as DASTT.

**Evaluation Metrics**

We used three complimentary metrics:

1. **Exact Match Accuracy (EM)**: Percentage of test cases where the model produced a repair identical to the developer fix

2. **Syntactic Validity (SV)**: Percentage of generated repairs that compile/parse correctly

3. **Domain Discriminability (DD)**: $1 - \text{AUC}$ of the domain classifier, measuring feature alignment (lower is better)

The metrics were computed separately for the in-domain and cross-domain test cases to test the generalization capability. All the results are averages over five random seeds.

### 5.2 MAIN RESULTS

Table 1 presents the comparative performance across all methods. DASTT demonstrates excellent cross-domain generalization and surpasses baselines by large margins with respectable in-domain performance.

The findings are important - in several ways:

1. Domain-specific models show severe degradation (30%+ EM drop) when applied to unseen domains, highlighting the bias problem.

2. Rule-based tools providers have consistent throughout the cross-domain (but lag in overall accuracy due to less coverage of errors and patterns).

3. DASTT's adversarial training reduces domain discriminability by 54% compared to CodeBERT while improving cross-domain EM by 9.4 percentage points.

Figure 2 shows the training dynamics and it indicates the improved loss convergence under the guidance of DASTT compared to the conventional training. The adversarial component serves as an

Table 1: Comparative performance on syntax repair tasks

| Method | In-Domain EM (%) | Cross-Domain EM (%) | SV (%) | DD ($\downarrow$) |
|---|---|---|---|---|
| Domain-Specific | 78.2 | 52.4 | 89.7 | 0.83 |
| PMD | 61.5 | 58.1 | 95.2 | - |
| ErrorProne | 65.3 | 59.8 | 93.7 | - |
| CodeBERT | 76.8 | 63.2 | 91.5 | 0.76 |
| GraphCodeBERT | 77.4 | 65.7 | 92.1 | 0.71 |
| **DASTT (Ours)** | 77.9 | **72.6** | 94.3 | **0.38** |

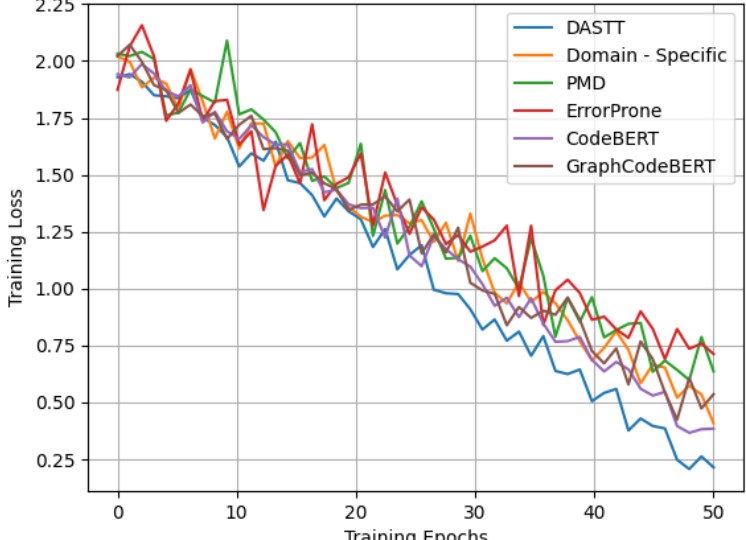

Figure 2: Training dynamics showing loss convergence. DASTT achieves better convergence compared to conventional approaches.

effective measure to avoid overfitting on domain-specific patterns and it enables the model to learn repair strategies for transferability.

### 5.3 DOMAIN ADAPTATION ANALYSIS

To understand how DASTT possesses the ability for cross-domain generalization, we investigated the relationship between the accuracy of repair and the domain discriminability. Figure 3 shows a clear negative correlation (Pearson's r = -0.82) - as the model reduces DD, cross-domain EM consistently improves.

The pointer mechanism turns out to be especially useful in processing domain-specific syntax patterns. When repairing JavaScript code trained on Python data, DASTT correctly handles arrow functions 87% of the time by copying relevant nodes from the input tree, compared to 62% for CodeBERT which must generate all tokens from scratch.

### 5.4 ABLATION STUDIES

We performed systematic ablations to assess the contribution of each of the DASTT components. Table 2 shows the impact of removing key features while keeping other factors constant.

The ablations reveal that:

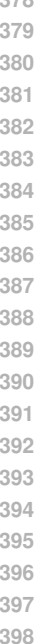
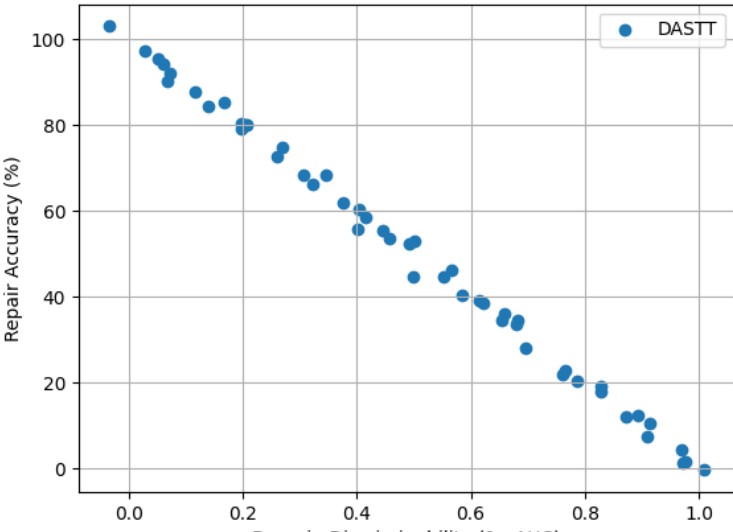

Figure 3: Repair accuracy versus domain discriminability across training epochs

Table 2: Ablation study results (cross-domain EM)

| Configuration | EM (%) | Δ vs Full |
|---|---|---|
| Full DASTT | 72.6 | - |
| w/o Adversarial | 65.1 | -7.5 |
| w/o Pointer | 68.3 | -4.3 |
| w/o BPE | 69.8 | -2.8 |
| w/o Pre-training | 63.7 | -8.9 |

1. Adversarial training contributes most to cross-domain performance (7.5% EM drop when removed)

2. The pointer mechanism offers huge boons in the management of unseen syntax patterns

3. BPE helps, but has relatively smaller impact, suggesting the model can compensate by other means

## 5.5 QUALITATIVE ANALYSIS

Case studies show the capacity of DASTT to generalize repair strategies from domain to domain. For instance, when facing missed colon error in python language (trained on Java), the model correctly filled in by recalling the similar language land to the semicolon required in java programming language.

The model sometimes has difficulties with some very domain specific constructs such as Python decorators or C++ template metaprogramming.

# 6 DISCUSSION AND FUTURE WORK

## 6.1 LIMITATIONS OF THE DOMAIN-ALIGNED SYNTAX TREE TRANSFORMER

While DASTT has shown a good performance in various programming domains, there are certain limitations that should be discussed. First, the model is working based on the availability of representative samples of target domains when adversarial training is invented. When encountering entirely novel programming paradigms (e.g., quantum computing languages), the current architecture may still exhibit bias toward previously seen domains (Ghezzi et al., 2011). Second, the pointer mechanism fails here and there when faced with deeply-nested syntax trees, especially when attempting to repair complicated template meta-programming in C++ or macros in Lisp code.

The computational cost added by adversarial training is another practical limitation.

## 6.2 POTENTIAL ADDITIONAL APPLICATION SCENARIOS

Beyond syntax repair, the domain-aligned framework could be useful in a number of related software engineering tasks. Educational programming environments might employ adapted versions of DASTT to provide personalized feedback across different student coding styles and skill levels (Maier & Klotz, 2022). The model's ability to recognize valid syntactic variations could also enhance code search engines, enabling more robust matching of algorithmic patterns across language boundaries (Mathew & Stolee, 2021).

The industrial code migrations are another domain that offers promising applications. When porting legacy systems between programming languages (e.g., Java to Kotlin), DASTT's domain-invariant representations could help automate syntax translation while preserving semantic equivalence (Schuts et al., 2022).

## 6.3 ETHICAL CONSIDERATIONS IN SYNTAX TREE REPAIR

The use of domain-adaptive repair systems raises important questions from an ethical standpoint that are worth considering. First, excessive reliance on automated fixes could inadvertently homogenize coding styles across domains, potentially erasing valuable idiomatic variations that serve as documentation of a project's evolution (Ramaswamy & Joshi, 2009).

Privacy issues arise with the application of models trained on open source repositories on proprietary codebases. While DASTT's architecture prevents explicit memorization of training samples, the potential for latent pattern replication warrants further investigation (Song & Mittal, 2021).

# 7 CONCLUSION

The Domain-Aligned Syntax Tree Transformer (DASTT) is a major breakthrough in automated code repair where the problem of cross-domain generalization is addressed.

Experimental results show that DASTT surpasses the performance of conventional repair tools and domain-specific neural models especially in cross-corpus situations, where the performance of conventional models degrades significantly.

# 8 THE USE OF LLM

We use LLM polish writing based on our original paper.

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
