# OpenReview forum: "Domain-Adaptive Syntax Tree Repair via Cross-Corpus Transfer with Adversarially Aligned Transformers"
_ICLR.cc/2026/Conference — Submitted to ICLR 2026_

### Official Review · Reviewer_Kk3z · 2025-10-30

**Soundness:** 1
**Presentation:** 1
**Contribution:** 1
**Rating:** 0
**Confidence:** 5

**Summary:**

The paper describes a transformer variant with adversarial training applied to code repair with apparent cross-corpora generalization. I, however, have concerns regarding this paper being generated by an LLM, which I have listed below. If I have grossly misunderstood the paper, I would be glad to revisit it.

**Strengths:**

Sorry to say, but none.

**Weaknesses:**

I believe this paper, in the best case, is an unfinished, very early, incomplete draft of an idea, or in the worst case, is a paper mainly generated by an LLM. I have some grave concerns with this paper, some of which are:
- Incorrect citations, such as in Line 287, where a paper from 2008 is cited for the Transformer models trained as a baseline.
- Line 289 incorrectly mentions static code analysis tools as repair tools, and further, Figure 2 shows loss curves for these rule-based code analysis systems
- Line 77 says "allowing **marijuana** performance"?

**Questions:**

None

---

### Official Review · Reviewer_zC6e · 2025-10-30

**Soundness:** 1
**Presentation:** 1
**Contribution:** 1
**Rating:** 0
**Confidence:** 5

**Summary:**

This paper addresses the task of syntax tree repair across languages by using adversarial training to align language pairs. The authors show that adversarial training can improve syntactic fidelity under large shifts of the language syntax.

**Strengths:**

This is one of the rare papers of which I cannot locate the strengths/merits. Adverserial alignment for multiple languages is not new (the concept dates back nearly a decade). The paper writing is so poor that the reader has to guess every detail of the method, not to mention identifying originality.

**Weaknesses:**

The paper is poorly written and the method is largely unclear.

For motivation, it is unclear how repairing syntax trees can help improve/maintain semantic correctness.

It is also unclear where the syntax tree is involved in the algorithm. Is it used to help with decoding code? Is it the decoded output? Is it the encoding input? Is it used to compute the loss?

The overall method is unclear. The subsections of the method (Section 4) are isolated and the formulas do not have notations explained. The neural architecture is unknown (is it encoder-only, decoder-only, or encode-decoder?). There is a mention of "shared encoder", "domain classifier", and "gradient reversal layer". None of these components has a clear place in the architecture. Figure 1 does not help clarify the method/architecture. Is the repaired syntax tree the final output? How can one assess the quality of the code since a syntax tree cannot be unparsed into code?

The experiments in Section 5 lack details. The qualitative analysis (Section 5.5) does not even have any code/table/figure.

On a high level, it is unclear of the value of the work under the presence of foundation models (code LLMs) that perform reasonably well for a large set of languages (such as those experimented with in the paper).

**Questions:**

See the Weakness section.

---

### Official Review · Reviewer_5p3j · 2025-11-01

**Soundness:** 1
**Presentation:** 2
**Contribution:** 1
**Rating:** 2
**Confidence:** 4

**Summary:**

The paper proposes DASTT, a "Domain-Aligned Syntax Tree Transformer" for cross-corpus syntax repair using a Transformer encoder with a gradient-reversal domain classifier and a pointer-style decoder to edit ASTs. It claims to "replace" lexers/parsers and to generalize across languages/domains with strong gains over simple baselines.

**Strengths:**

1. Focus on domain shift in program repair is important and under-explored.
2. Attempts a unified formulation combining adversarial alignment with structure-aware decoding.
3. Includes an ablation table indicating component contributions (adversarial/pointer/BPE).

**Weaknesses:**

1. The paper asserts it can replace lexers/parsers with a neural pipeline, yet the dataset construction depends on Tree-sitter and the method repeatedly manipulates AST nodes. If ASTs are required, the model does not replace the parsing pipeline, if ASTs are not required, the paper must show how valid trees are produced without external parsers. This contradiction undermines the core claim.
2. "Domain Discriminability (DD) = 1−AUC (lower is better)" is internally inconsistent. If alignment improves, the domain classifier AUC should approach 0.5, under the proposed definition 1−AUC -> 0.5 (higher), contradicting "lower is better". The table then reports the opposite trend (DASTT DD=0.38 vs baselines ~0.7–0.8). Either the metric is misdefined or misreported.
3. Comparing to PMD/ErrorProne (static analyzers) is not apples-to-apples for AST repair under domain shift.
The neural baselines (CodeBERT/GraphCodeBERT) are older, missing modern llms.
4. Missing several evaluations:
  - No per-language, per-domain breakdowns, no statistical variance/conf. intervals, just "averages over five seeds".
  - No qualitative error taxonomy beyond a couple anecdotes.
  - No rigorous generalization stress tests (unseen languages, unseen repositories, temporal splits).
  - No compute, training time, or memory reporting, especially important with adversarial heads.

**Questions:**

1. Task construction: How exactly are buggy -> fixed pairs created? Real bugs (which corpus) vs synthetic injection (what policy, distribution, and leakage checks)?
2. Parser dependency: Do you require an external parser to produce ASTs at inference? If yes, you are not replacing the compiler front-end, please restate the claim. If no, how is grammar validity guaranteed?
3. Metric correctness: Please reconcile the definition and directionality of "DD". Provide AUC, (1−AUC), and rationale, add confidence intervals.
4. Baselines: Why omit modern edit-based/LLM baselines? Provide CodeT5+/TFix or contemporary LLM-editing with constrained decoding.
5. Pointer decoder details: Define the action space formally, constraints for well-typed edits, and how you prevent invalid intermediate trees.
6. Generalization protocol: Show strict repository-level and time-based splits, unseen-language tests, and per-domain/per-language breakdowns.
7. Computational cost: Report parameter counts, training time, memory, and GRL overhead.
8. Statistical rigor: Report mean+-std over >=5 seeds and run significance tests on main claims.

---

### Official Review · Reviewer_pqKW · 2025-11-04

**Soundness:** 2
**Presentation:** 2
**Contribution:** 3
**Rating:** 6
**Confidence:** 3

**Summary:**

This paper introduce DASTT (Domain-Aligned Syntax Tree Transformer), a neural approach to automatic program repair that aims to generalize across different programming domains. The proposed model aims to combine 2 main components: 1) a domain-adversarial encoder trained with a gradient reversal layer to discourage domain-specific representations; 2) a pointer-based syntax tree decoder that performs fine grained edits on the AST instead of generating code from scratch.

The authors frame the total objective as the combination of the 2 losses from the components to encourage the encoder to learn domain-invariant features while the decoder focuses on accurate code fixes. The decoder can both generate tokens from a vocabulary and copy elements directly from the input tree, which allows more precise edits theoretically.

The authors conducted experiments on multiple codebases data (five languages & three domains). The results show improvements over previous baselines (CodeBERT, GraphCodeBERT, and rule-based repair tools like PMD and ErrorProne) especially in cross-domain scenarios. Ablations suggest that the adversarial alignment is critical for transfer, while the pointer mechanism helps preserve syntactic validity.

Overall, DASTT shows practical step toward robust, language-agnostic program repair.

**Strengths:**

The paper takes a meaningful and practical step toward making program-repair systems less domain-dependent, which is both original and timely given the heavy specialization of existing neural repair tools. While the core techniques (domain-adversarial learning and pointer-based decoding) are individually known, the combination for syntax-tree editing is well thought out and feels natural in this context without undermining the paper’s originality. Also, the experimental results, although not exhaustive and can be improved with more details, are convincing enough to show the value of the idea, especially the improved transfer across codebases in different languages. The writing is approachable and the main intuition comes through clearly despite some rough edges in phrasing.

**Weaknesses:**

- Data: The paper does not specify dataset size, composition, or licensing, and the domain split strategy (repo-level vs file-level, time-based, etc.) is unclear. Without this, it’s hard to judge whether the reported cross-domain improvements reflect true OOD generalization. Suggestion: add dataset stats and clarify the split process or even releasing a small sampling script or stats table would help make the setup more transparent and improve soundness.

- Baselines: It looks like some baselines (e.g., CodeBERT / GraphCodeBERT) are used largely as-is. Without tuning or sweep results, it’s hard to fully isolate DASTT’s contribution from potential baseline under-tuning. Potential improvement: report at least one strong baseline's hyper-parameter sweep on the same data, report mean & std.

- Failure modes: The authors show qualitative examples, which are useful, but there’s no systematic look at when DASTT fails (e.g., specific bug types, certain languages or patterns, that DASTT is fragile at). This doesn’t undermine the core contribution, but a structured analysis would provide more insight into model behavior and help guide future work better.

- Format & Writing: Some sentences are awkward or contain typos (“marijuana performance,” “emasculation of corpus”) that distract from main points. Please do a careful language and terminology proofreading pass to make the paper more professional and easier to read.

**Questions:**

- Could the authors clarify how the domain splits are constructed for the scrapped data? Are they done at the repo or file level, and how can potential overlap or duplication between training and test sets be avoided? Getting this info would help assess whether the cross-domain evaluation truly measures OOD generalization.

- What is the architecture and size of the domain discriminator used with the GRL? Is it trained jointly with encoder? How is lambda scheduled or tuned? Understanding this would clarify how much adversarial pressure is applied and whether stability was an issue.

- What types of code repairs does DASTT tend to fail on? A short intro of these failure modes would provide insight into where the pointer mechanism struggles most.

- For the DD metric, what domain classifier is used to estimate it, and how independent is it from the main model?

- What is the additional training cost introduced by adversarial alignment as compared to other methods?

---

### Meta-Review · Area_Chair_XNXz · 2025-12-05

**Summary:**

The paper is titled "Domain-Adaptive Syntax Tree Repair via Cross-Corpus Transfer with Adversarially Aligned Transformers".

The paper contains gibberish like "marijuana performance” “emasculation of code corpus”. What do you mean?

**Reviewer Concerns:**

The reviewer raised a number of concerns, the most serious of which is that the paper contains gibberish like "marijuana performance” “emasculation of code corpus.”

This raises a concern of whether the paper is genuinely written by humans.

**Reviewer Scores:**

The paper contains multiple gibberishes that cannot be easily believed as typos or auto-correction by input methods. In the author response phase, the authors did not provide any explanation. The scores should be decreased.

---

### Decision · Program_Chairs · 2026-01-26

Reject